# Wave-Encoded Model-Based Deep Learning for Highly Accelerated Imaging with Joint Reconstruction

**DOI:** 10.3390/bioengineering9120736

**Published:** 2022-11-29

**Authors:** Jaejin Cho, Borjan Gagoski, Tae Hyung Kim, Qiyuan Tian, Robert Frost, Itthi Chatnuntawech, Berkin Bilgic

**Affiliations:** 1Athinoula A. Martinos Center for Biomedical Imaging, Massachusetts General Hospital, Boston, MA 02129, USA; 2Department of Radiology, Harvard Medical School, Boston, MA 02115, USA; 3Fetal-Neonatal Neuroimaging & Developmental Science Center, Boston Children’s Hospital, Boston, MA 02115, USA; 4Department of Computer Engineering, Hongik University, Seoul 04066, Republic of Korea; 5National Nanotechnology Center, Khlong Nueng, Klong Luang, Pathum Thani 12120, Thailand; 6Harvard/MIT Health Sciences and Technology, Cambridge, MA 02139, USA

**Keywords:** parameter mapping, model-based deep learning, wave-encoding, wave-MoDL

## Abstract

A recently introduced model-based deep learning (MoDL) technique successfully incorporates convolutional neural network (CNN)-based regularizers into physics-based parallel imaging reconstruction using a small number of network parameters. Wave-controlled aliasing in parallel imaging (CAIPI) is an emerging parallel imaging method that accelerates imaging acquisition by employing sinusoidal gradients in the phase- and slice/partition-encoding directions during the readout to take better advantage of 3D coil sensitivity profiles. We propose wave-encoded MoDL (wave-MoDL) combining the wave-encoding strategy with unrolled network constraints for highly accelerated 3D imaging while enforcing data consistency. We extend wave-MoDL to reconstruct multicontrast data with CAIPI sampling patterns to leverage similarity between multiple images to improve the reconstruction quality. We further exploit this to enable rapid quantitative imaging using an interleaved look-locker acquisition sequence with T_2_ preparation pulse (3D-QALAS). Wave-MoDL enables a 40 s MPRAGE acquisition at 1 mm resolution at 16-fold acceleration. For quantitative imaging, wave-MoDL permits a 1:50 min acquisition for T_1_, T_2_, and proton density mapping at 1 mm resolution at 12-fold acceleration, from which contrast-weighted images can be synthesized as well. In conclusion, wave-MoDL allows rapid MR acquisition and high-fidelity image reconstruction and may facilitate clinical and neuroscientific applications by incorporating unrolled neural networks into wave-CAIPI reconstruction.

## 1. Introduction

MRI has been widely used to provide structural and physiological images; however, its efficiency has been limited by an inherent tradeoff between scan time, resolution, and signal-to-noise ratio (SNR) [1]. MRI repeatedly excites and collects the signal while encoding the signal to visualize the acquired data as images, which often takes a long scan time. To mitigate the encoding burden of MRI scans, parallel imaging (PI) techniques have been developed to accelerate various sequences through the use of multiple receiver coils for image encoding. Sensitivity encoding (SENSE) [2] and generalized auto-calibrating partially parallel acquisition (GRAPPA) [3] have been widely used to reconstruct data in the image domain and *k*-space, respectively, and the recent ESPIRiT technique [4] has bridged the gap between these two domains. A number of techniques have been proposed to improve the conditioning of PI acquisition to enable higher accelerations. Controlled aliasing in parallel imaging (CAIPI) [5] modifies the appearance of aliasing artifacts during the acquisition to improve the subsequent PI reconstruction for multislice imaging. Application of interslice shifts to three-dimensional (3D) imaging forms the basis of 2D-CAIPI [6], wherein the phase ($k_{y}$) and partition ($k_{z}$) encoding positions are modified to shift the spatial aliasing pattern to reduce aliasing and better exploit the variations in coil sensitivities.

Wave-CAIPI is a more recent controlled aliasing method that can further reduce noise amplification and aliasing artifacts in highly accelerated acquisitions [7,8]. It employs extra sinusoidal gradient modulations in the phase- and the partition-encoding directions during the readout to better harness coil sensitivity variations in all three dimensions and achieve higher accelerations. Wave-encoding also incorporates 2D-CAIPI interslice shifts to improve the PI conditioning [6] and has found applications in highly accelerated gradient-echo (GE), MPRAGE, fast spin-echo, and echo-planar imaging acquisitions [8,9,10,11,12].

Compressed sensing (CS) and low-rank reconstruction using annihilation filters further improved the MR reconstruction [13,14,15,16,17,18]. In the last decade, combined PI and CS techniques have resulted in substantial improvements in acquisition speed and image quality. Although the PI–CS combination can achieve state-of-the-art performance [17,19,20,21,22], designing effective regularization schemes and tuning hyperparameters are nontrivial, and it often requires a relatively long reconstruction time.

Recently, image reconstruction with deep learning approaches has been explored [23,24,25] to overcome the hurdles faced by existing reconstruction techniques such as long reconstruction time, residual artifacts at high acceleration factor, and oversmoothness. A few studies have shown deep learning methods outperforming conventional regularization- and/or optimization-based techniques in various applications. PI reconstruction has been implemented with multilayer perceptron [23] and the variational network incorporating the unrolled network during conjugate gradient updates [24]. Some of the studies trained neural networks in *k*-space to exploit the features in the spatial frequency domain [25,26,27,28], as many PI techniques have been implemented in *k*-space. In recent works, the wave-encoding strategy was successfully combined with the variational network [29] and scan-specific image reconstruction [30,31] to provide high-quality images at high acceleration.

Recently developed model-based deep learning (MoDL) improves image reconstruction by leveraging an unrolled convolutional neural network (CNN) and PI forward model to help denoise and unalias undersampled data [32]. MoDL consists of unrolled CNN and data consistency blocks and updates the image by iteratively passing the data through both blocks. The sharing of network parameters across iterations enables MoDL to keep the number of learned parameters decoupled from the number of iterations, thus providing good convergence without increasing the number of trainable parameters. Such smaller number of trainable parameters also translates to significantly reduced training data demands, which is particularly attractive for data-scarce medical-imaging applications. MoDL has been further applied to multishot diffusion-weighted echo-planar imaging [33]. This work successfully replaced the multishot sensitivity-encoded diffusion data recovery algorithm using structured low-rank matrix completion (MUSSELS) [18] with the hybrid MoDL-MUSSELS approach which employs CNNs in both *k*-space and image domains. MoDL-MUSSELS was able to yield reconstructions that are comparable to state-of-the-art methods while offering several orders of magnitude reduction in run time.

Quantitative MRI estimates tissue parameters quantitatively, enabling the detection of, e.g., microstructural processes related to tissue remodeling in aging and neurological diseases [34]. Advanced image reconstruction could benefit high-resolution quantitative imaging as well, which often requires lengthy acquisitions because a multidimensional signal space needs to be sampled to enable parameter quantification. The 3D-QuAntification using an interleaved Look-Locker Acquisition Sequence with T_2_ preparation pulse (3D-QALAS) acquires high-resolution images with five different contrasts and enables simultaneous T_1_, T_2_, and proton density (PD) parameter mapping [35,36,37]. However, encoding limitations stemming from multicontrast sampling at high resolution substantially lengthen 3D-QALAS acquisitions, e.g., 11 min for 1 mm iso resolution at *R* = twofold acceleration [38]. The combination of PI with CS has recently enabled a 6 min scan at *R* = 3.8 [38]. Unfortunately, pushing the acceleration further is hampered by *g*-factor penalty and intrinsic SNR limitations.

In this study, we propose to combine wave-encoding and MoDL synergistically and demonstrate a rapid, 16-fold accelerated MPRAGE acquisition with high image quality. We further propose to expand our earlier work presented as an abstract [39] by using wave-MoDL to enable highly accelerated, high-resolution multiecho and multicontrast imaging with joint reconstruction. We first demonstrate the application of this in multiecho MPRAGE (MEMPRAGE) [40] imaging employed in the Human Connectome Project (HCP) at 0.8 mm isotropic resolution and jointly reconstruct four echoes at *R* = ninefold acceleration. Moreover, we make use of controlled aliasing by shifting the sampling patterns across the echoes, thereby increasing the collective *k*-space coverage. Finally, we demonstrate the ability to perform joint multicontrast reconstruction with wave-MoDL on the 3D-QALAS sequence. We pushed the acceleration to *R* = 12-fold to provide a 1:50 min comprehensive quantitative exam by jointly reconstructing multicontrast 3D-QALAS images with wave-MoDL, thereby minimizing the *g*-factor loss using wave-encoding and boosting SNR with MoDL. From this rapid 1:50 min 3D-QALAS scan, wave-MoDL allows for the estimation of high-quality T_1_, T_2_, and PD parameter maps, which can be used for synthesizing standard contrast-weighted images that lend themselves to clinical reads or subsequent analysis using image analysis software (e.g., FreeSurfer) [41,42].

## 2. Theory

### 2.1. Wave-CAIPI

Wave-encoding employs additional sinusoidal gradients during the readout to take better advantage of 3D coil sensitivity profiles [7,8,10,43]. Wave-encoding modulates the phase during the readout and incurs a corkscrew trajectory in *k*-space. This spreads the aliased signal into the readout direction, thereby employing an additional sensitivity profile for image reconstruction. The wave-encoded signal, *s*, can be explained using the following equation.
(1)$$\begin{array}{cc} s(t) & =\int_{x,y,z}m\left(x,y,z\right)\cdot e^{-2\pi i\left(k_{x}\left(t\right)x+k_{y}y+k_{z}z\right)}\cdot e^{-\gamma i\int_{0}^{t}\left(g_{y}\left(\tau \right)y+g_{z}\left(\tau \right)z\right)d\tau }dxdydz\\ \; & =\int_{x,y,z}m\left(x,y,z\right)\cdot e^{-2\pi i\left(k_{x}\left(t\right)\left(x+\frac{\gamma \int_{0}^{t}\left(g_{y}\left(\tau \right)y+g_{z}\left(\tau \right)z\right)d\tau }{2\pi k_{x}\left(t\right)}\right)\right)}\cdot e^{-2\pi i\left(k_{y}y+k_{z}z\right)}dxdydz \end{array}$$
where *m* is the magnetization, γ is the gyromagnetic ratio, and *g* is the time-varying sinusoidal wave gradient. Voxel spreading by wave-encoding is a function of readout time and the position in the *y*- and *z*-dimensions ($k_{x}$-*y*-*z* hybrid domain). The reconstruction of standard wave-CAIPI is described as follows:(2)$$\begin{array}{cc} m & =\underset{m}{\mathrm{argmin}}{∥WF_{y}PF_{x}Cm-s∥}_{2}^{2}\\ \; & =\underset{m}{\mathrm{argmin}}{∥A(m)-s∥}_{2}^{2} \end{array}$$
where m is the reconstructed image, W is the subsampling mask, F is the Fourier transform operator, P is the wave point spread function in the $k_{x}$-*y*-*z* hybrid domain (corresponding to $\int_{0}^{t}\left(g_{y}\left(\tau \right)y+g_{z}\left(\tau \right)z\right)d\tau$ in Equation (Equation 1)), C is the coil sensitivity map, and s is the subsampled *k*-space data.

### 2.2. 3D-QALAS

Figure 1a shows the sequence diagram of 3D-QALAS that enables high-resolution and simultaneous T_1_, T_2_, and PD parameter mapping. The 3D-QALAS is based on a multiacquisition 3D gradient echo sequence, with five acquisitions equally spaced in time, interleaved with a T_2_-preparation pulse and an inversion pulse. The first readout succeeds a T_2_-preparation module, and the remaining four readouts follow an inversion pulse that captures T_1_ dynamics. The estimated parameter maps will provide the ability to synthesize standard contrast-weighted images, thus allowing the sequence to be used in a more traditional way for clinical reads. We used a standalone version of SyMRI (SyntheticMR, Linköping, Sweden) to generate quantitative parameter maps and synthetic images.

## 3. Methods

### 3.1. Wave-MoDL

We propose wave-MoDL incorporating the wave-encoding strategy into MoDL image reconstruction to improve the PI condition and high-resolution images with high image quality. For single-image contrast, wave-MoDL reconstruction can be described as follows.
(3)$$\begin{array}{cc} m & =\underset{m}{\mathrm{argmin}}{∥WF_{y}PF_{x}Cm-s∥}_{2}^{2}+\lambda_{1}{∥N_{k}\left(m\right)∥}_{2}^{2}+\lambda_{2}{∥N_{i}\left(m\right)∥}_{2}^{2}\\ \; & =\underset{m}{\mathrm{argmin}}{∥A(m)-s∥}_{2}^{2}+\lambda_{1}{∥N_{k}\left(m\right)∥}_{2}^{2}+\lambda_{2}{∥N_{i}\left(m\right)∥}_{2}^{2} \end{array}$$
where $N_{k}$ and $N_{i}$ represent residual CNNs in the *k*- and image-space, respectively. Figure 1b shows the proposed network architecture of wave-MoDL reconstruction for single/multicontrast images. The unrolled CNN block captures image features to help the image reconstruction. We use unrolled CNNs to constrain the reconstruction in both the image- and *k*-space, as it was shown previously that applying constraints in both domains improves the performance of the entire network [26,33]. The data consistency (DC) block enforces consistency with measured wave-encoded signals where wave point spread function (wave-PSF) input is required to utilize the known forward model. At the DC block, the resulting quadratic subproblem was solved using conjugate gradient (CG) optimization. The residual CNN, N, can be explained by CNN, D, and the skip connection, N(m)=m−D(m). We used the alternating recursive minimization-based solution as described in [32,33], by which the network is unrolled. By substituting $\eta_{n+1}=D_{k}\left(m_{n+1}\right)$ and $\zeta_{n+1}=D_{i}\left(m_{n+1}\right)$, where $D_{k}$ and $D_{i}$ represent CNNs in the *k*- and image-space, respectively, an alternating minimization-based solution can be described as follows.
(4)$$\begin{array}{cc} m_{n+1} & ={\left(A^{H}A+\lambda_{1}I+\lambda_{2}I\right)}^{-1}\left(A^{H}s+\lambda_{1}\eta_{n}+\lambda_{2}\zeta_{n}\right)\\ \eta_{n+1} & =D_{k}\left(m_{n+1}\right)\\ \zeta_{n+1} & =D_{i}\left(m_{n+1}\right) \end{array}$$
where *n* is the iteration number and *H* is the Hermitian transpose.

### 3.2. Wave-MoDL for Multicontrast Image Reconstruction

We further extended single-contrast wave-MoDL to reconstruct multicontrast images, m, from Equations (2) and (3), as follows.
(5)$$\begin{array}{cc} m & =\underset{m}{\mathrm{argmin}}\sum_{l}^{L}{∥W_{l}F_{y}PF_{x}Cm_{l}-s_{l}∥}_{2}^{2}+\lambda_{1}{∥N_{k}\left(m\right)∥}_{2}^{2}+\lambda_{2}{∥N_{i}\left(m\right)∥}_{2}^{2}\\ \; & =\underset{m}{\mathrm{argmin}}\sum_{l}^{L}{∥A_{l}\left(m_{l}\right)-s_{l}∥}_{2}^{2}+\lambda_{1}{∥N_{k}\left(m\right)∥}_{2}^{2}+\lambda_{2}{∥N_{i}\left(m\right)∥}_{2}^{2} \end{array}$$
where *L* is the number of image contrasts, $W_{l}$ is the subsampling mask for the *l*-th contrast, $m_{l}$ is the *l*-th contrast image, and $s_{l}$ is the *l*-th subsampled *k*-space data, respectively. We concatenate $m_{l}$ and create the variable m comprising all contrasts, and provide this as the input channel of residual CNN, N. CAIPI sampling patterns were applied across the contrasts to improve the reconstruction by increasing the collective *k*-space coverage [44].

### 3.3. Experiments

We trained, validated, and tested the wave-MoDL using three different databases acquired on a 3T Siemens Prisma system equipped with a 32-channel head receive array. Table 1 shows the acquisition parameters of the databases and the networks. Figure 2 shows the *g*-factor analyses of SENSE and wave-CAIPI, linear reconstructions with neither network nor regularization, at the target acceleration for each database. We separated the 3D data into slice groups and trained on sets of aliasing slices to address the GPU memory constraint. The number of slices of an input 3D batch image is equal to the acceleration factor in the slice-encoding direction. For example, at $R_{z}=3$, a batch contains three slices that are aliasing on each other slice and need to be unaliased. Multiple contrast images were concatenated in the input channel dimension of the network. The wave-MoDL network updates the results during 10 outer iterations and takes 10 conjugate gradients per iteration in the data consistency layers. The unrolled CNN has five hidden layers consisting of a 24-depth filter with a leaky ReLU activation per layer, and all network parameters were zero-initialized. Coil sensitivity maps were calculated from external 3D low-resolution gradient-echo-based reference scans using ESPIRiT [4]. Example code can be found at https://github.com/jaejin-cho/wave-modl (accessed on 4 April 2022).

#### 3.3.1. MPRAGE Database

We acquired fully sampled T_1_-MPRAGE data on 10 healthy subjects at 1 mm isotropic resolution to generate the database. We used a split of 8/1/1 subjects for training/validating/testing the wave-MoDL network. The network had access to 384 slice groups during the training, which should be a sufficient database size considering the number of network parameters as detailed in [32]. We retrospectively subsampled the data at the acceleration factor of *R* = 4 × 4, where wave-encodings were applied in both $k_{y}$ and $k_{z}$ directions with 8.8 mT/m of G_max_ and 11 cycles [10]. The receiver bandwidth was 200 Hz/pixel. From the *g*-factor analysis in Figure 2, we expect 2.5-fold averaged *g*-factor gain and 5.4-fold maximum *g*-factor gain by incorporating wave-encoding strategy with respect to SENSE image reconstruction at *R* = 4 × 4.

#### 3.3.2. MEMPRAGE Database

*k*-space data from four-echo MEMPRAGE scans were collected at the Massachusetts General Hospital HCP-Aging site [45]. The database consists of 30 subjects at 0.8 mm isotropic resolution and reduction factor 2. The network had access to 1892 slice groups (22 subjects) during the training. To create a database comprising high-quality reference images, we reconstructed images using GRAPPA on each of the four different echoes separately [3,4]. We retrospectively subsampled the data to *R* = 3 × 3 and incorporated CAIPI sampling patterns to use complementary information for the reconstruction across the multiechos. Wave-encoding was applied in both $k_{y}$ and $k_{z}$ directions with 9.626 mT/m of G_max_ and four cycles at 744 Hz/pixel of receiver bandwidth. Incorporating wave-encoding reduces the average *g*-factor by 13% and the maximum *g*-factor by 43% with respect to SENSE at *R* = 3 × 3, as shown in Figure 2.

#### 3.3.3. 3D-QALAS Database

We scanned 10 healthy subjects using the 3D QALAS sequence (Figure 1a) at 1 mm isotropic resolution and reduction factor of 2. To avoid prohibitively long acquisition times, data were acquired at *R* = 2 to limit the scan time to 11 min. High-SNR reference images were reconstructed using GRAPPA [3] for the five different contrasts separately. To train/validate/test the network, 8/1/1 subjects were used, respectively. The network had access to 512 slice groups during the training. The data were retrospectively subsampled to *R* = 4 × 3, corresponding to a 1:50 min quantitative exam, where 5-cycle cosine and sine wave-encodings were applied in both $k_{y}$ and $k_{z}$ directions with 16.5 mT/m of G_max_ at 347 Hz/pixel bandwidth. To take into account the signal intensity differences between the contrasts, the contrast images were weighted by global scaling factors [3.26, 2.36, 1.57, 1.12, 1], calculated by the L2 signal norm ratio of each contrast image in the training database. This allowed each contrast to contribute to the loss function in similar amounts. We applied CAIPI sampling patterns across the multiple contrasts to use the complementary *k*-space information from each other echo in the reconstructions using MoDL and wave-MoDL. A fixed sampling pattern was applied to the five different contrasts for SENSE and wave-CAIPI, since these algorithms reconstruct each image independently. Employing wave gradients reduces the average *g*-factor by 20% and improves the maximum *g*-factor by 2.5-fold with respect to SENSE, as shown in Figure 2.

## 4. Results

### 4.1. MPRAGE at R = 4 × 4

Figure 3 shows the reconstruction results at *R* = 4 × 4 on the MPRAGE data. SENSE suffered from residual folding artifacts and noise amplification. MoDL significantly mitigated the noise amplification and reduced NRMSE by 1.78-fold with respect to SENSE, but still suffered from folding artifacts (as shown by the arrows). Wave-CAIPI reduces NRMSE by 1.96-fold with respect to SENSE. Wave-MoDL further reduced the noise amplification and decreased NRMSE to 7.20%, which is a 1.92-fold improvement with respect to wave-CAIPI, and a 2.10-fold improvement over standard MoDL. Wave-MoDL thus provided high-quality image reconstruction at *R* = 4 × 4-fold acceleration, allowing the acquisition of whole-brain structural data in 40 seconds at 1 mm isotropic resolution.

### 4.2. MEMPRAGE at R = 3 × 3

Figure 4 shows the reconstruction results of MEMPRAGE at 0.8 mm isotropic voxels and *R* = 3 × 3-fold acceleration. The images shown are the root mean squared (RMS) combination of the four echoes. CAIPI sampling patterns across the echoes were applied to MoDL and wave-MoDL for better use of the multiecho information, which in turn improves the reconstruction. Moreover, fixed sampling patterns were applied in SENSE and wave-CAIPI reconstruction since these algorithms independently reconstruct each echo image. Appendix A shows the sampling patterns and each echo image before the combination. SENSE suffered from high noise amplification. Although MoDL mitigated the noise amplification, the part of the folding artifacts still remains (as pointed out by the arrows). Since wave-encoding was limited by a high receiver bandwidth [12], and wave-CAIPI also suffered from some noise amplification and resulted in even higher NRMSE with respect to Cartesian MoDL. Wave-MoDL shows improved reconstructions and reduced NRMSE to 11.18%. The image reconstruction at *R* = 3 × 2 on the same database is shown in Appendix A.

### 4.3. 3D-QALAS at R = 4 × 3

Figure 5 shows the multicontrast images of the five 3D-QALAS readouts, reconstructed using SENSE, MoDL, wave-CAIPI, and wave-MoDL at *R* = 4 × 3-fold acceleration. SENSE suffered from aliasing artifacts and noise amplification, while MoDL mitigated noise amplification and reduced the NRMSE significantly. Wave-CAIPI also improved the reconstruction and reduced the NRMSE to 9.15%. Wave-MoDL further reduced NRMSE to 6.49%, which is a 1.4-fold improvement with respect to both standard Cartesian MoDL and wave-CAIPI.

Figure 6 shows the T_1_, T_2_, and PD quantification results. To evaluate the accuracy, we calculated averaged values and standard deviation in white matter (WM, brown) and gray matter (GM, purple) segmented by FreeSurfer [41,42]. In the last column, we selected 50 brain regions, each including 5×5×5 voxels in WM and GM, and plotted the quantified values using the wave-MoDL results over the reference *R* = 2 acquisition. The plotted graphs demonstrate that the estimated T_1_, T_2_, and PD values are well aligned with the reference QALAS acquisition. In GM, the estimated T_1_, T_2_, and PD were 1214 ± 477, 82 ± 50, and 76 ± 10, respectively, whereas the reference T_1_, T_2_, and PD were 1190 ± 474, 83 ± 61, and 76 ± 9, respectively. In WM, the estimated T_1_, T_2_, and PD were 839 ± 162, 67 ± 10, and 69 ± 6, respectively, whereas the reference T_1_, T_2_, and PD were 804 ± 162, 66 ± 9, and 68 ± 6, respectively. R2 metrics were 0.956, 0.965, and 0.926 in T_1_, T_2_, and PD maps, respectively. As shown in Figure 5, wave-MoDL mitigated the noise amplification in the parameter maps as well.

On the QALAS database, using NVidia Quadro RTX 5000, the whole brain image reconstructions took 7:35, 8:11, 14:35, and 15:26 min for SENSE, MoDL, wave-CAIPI, and wave-MoDL, respectively. Most of the image reconstruction time came from the DC layers, whereas unrolled networks took an additional 36 and 51 s in Cartesian and wave-encodings.

## 5. Discussion

We introduced the wave-MoDL acquisition/reconstruction strategy that synergistically combined wave-encoding with unrolled deep learning reconstruction and showed markedly improved image quality at high acceleration rates. In vivo experiments show its ability to provide anatomical images in 40 s using a 16-fold accelerated MPRAGE scan, 1:30 min submillimeter imaging with high fidelity with an MEMPRAGE, and quantitative images using a 1:50 min 3D-QALAS acquisition. Though the current implementation is not able to use the information from adjacent slices due to the limited GPU memory, it significantly reduces the memory footprint and facilitates training with high channel count data.

We employed CAIPI sampling patterns across the multiple echoes/contrasts to improve the reconstruction by leveraging the similarity of the multiple images [44]. To evaluate the efficiency of multicontrast information use, we reconstructed the images with and without CAIPI sampling patterns on the QALAS data, and trained five independent wave-MoDL networks to reconstruct each image separately, as shown in Figure 7. This figure demonstrates that the use of CAIPI sampling patterns helps reconstruct the images using shared information and reduces NRMSE from 7.38% to 6.49%. Wave-MoDL for multicontrast reconstruction without the CAIPI sampling patterns has comparable NRMSE with respect to independently trained networks for each echo while reducing the number of network parameters by ~5-fold. Wave-MoDL with CAIPI sampling patterns across contrasts improved the reconstruction while using a small number of network parameters, as the network can obtain complementary information from each echo image.

The 3D-QALAS can estimate the T_1_, T_2_, and proton density parameter maps; thus, additional contrast-weighted images can be synthesized from the quantitative parameter maps. Figure 8 shows the synthesized T_1_w, T_2_w, and FLAIR images using the quantified T_1_, T_2_, and PD maps obtained using the SyMRI software tool. The results demonstrate that wave-MoDL can mitigate the noise amplification and provide synthetic images that match well to the reference *R* = twofold accelerated data with higher fidelity than SENSE, MoDL, and wave-CAIPI. The synthesized PDw, double inversion recovery (DIR), and phase-sensitive inversion recovery (PSIR) images are shown in Appendix A.

The proposed wave-MoDL reconstruction method uses unrolled network constraints and successfully improved image quality at high acceleration rates. In Appendix A, on the MEMPRAGE database, we explored other image reconstruction constraints, low-rank property over multiple echoes with the 3D wave-LORAKS strategy [11], and 2D U-net denoisers, of which the inputs are four-echo wave-CAIPI images. With the mean squared error, we trained two different U-net denoisers; one is a typical U-net and another one is a small U-net that has a similar number of network parameters to the one of wave-MoDL. Wave-LORAKS could improve both NRMSE and SSIM from the wave-CAIPI result, but is still noisier than wave-MoDL. Compared with wave-MoDL, the first U-net denoiser shows the best NRMSE on root mean squared images with 378 times more network parameters. The second U-net denoiser comprises 15% more network parameters and shows comparable NRMSE to wave-MoDL. However, SSIMs of the U-net denoisers are worse than the results using other constraints. Wave-MoDL presents the best SSIM and better NRMSE improvement per network parameter.

Because the networks were trained using databases that included only healthy subjects, the trained networks’ ability to generalize can be improved by incorporating data from pathological cases. Scan-specific models might help mitigate this concern on generalization to unseen pathologies [27,46]. Recent work on wave-spark could achieve 1.2-fold NRMSE reduction with respect to wave-CAIPI at *R* = 15 MPRAGE data [31], whereas our wave-MoDL could obtain 1.9-fold NRMSE reduction with respect to wave-CAIPI for *R* = 16-fold accelerated MPRAGE. This may suggest that database-trained models may provide higher performance gains than scan-specific approaches, albeit at the potential cost of reduced ability to generalize to unseen pathologies [47].

As we train the network at specific reduction factors and using CAIPI sampling patterns, the benefits including the NRMSE reduction can be decreased at different accelerations and/or CAIPI sampling patterns. In Figure 9, we reconstructed HCP images at *R* = 3 × 3 using wave-CAIPI, wave-MoDL trained with data at the difference acceleration factor of *R* = 3 × 2, and wave-MoDL trained with data at *R* = 3 × 3. Even though the networks of wave-MoDL were trained at a different reduction factor (*R* = 3 × 2) and sampling patterns, the networks could denoise the image and reduce NRMSE with respect to wave-CAIPI. However, it also generated amplified folding artifacts, as pointed out by arrows, which are not present in the standard wave-CAIPI reconstructions or the wave-MoDL results trained with *R* = 3 × 3 data. The networks trained at the same reduction factor (*R* = 3 × 3) and sampling pattern show the best performance. Fine-tuning the network with a couple of subject data might be required to use the network trained at a different reduction factor or different sampling pattern.

## 6. Conclusions

We showed that our proposed wave-MoDL method enables highly accelerated MR scans that improve image fidelity and acquisition speed. Its application was demonstrated in MPRAGE, MEMPRAGE, and 3D-QALAS sequences. Joint-wave-MoDL was introduced to improve the reconstruction by employing the information from other echoes/contrasts using CAIPI sampling patterns. Extension of joint image reconstruction using wave-MoDL to 3D-QALAS enabled a high acceleration rate of *R* = 12 with a 32-channel coil array. This permitted a 1:50 min 3D-QALAS acquisition at 1 mm isotropic resolution and simultaneous T_1_, T_2_, and PD quantification as well as the synthesis of contrast-weighted images with high fidelity.

## Figures and Tables

**Figure 1 bioengineering-09-00736-f001:**
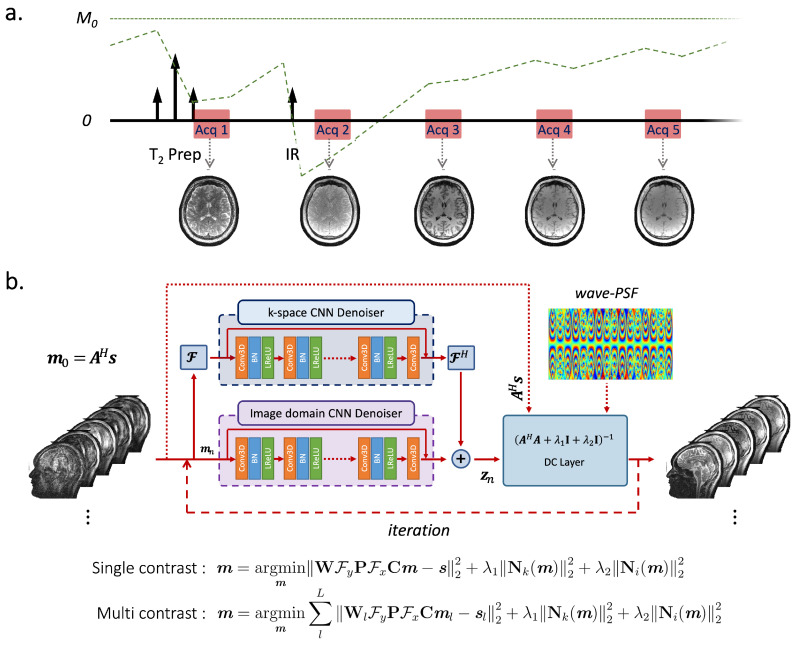
(**a**) 3D-QALAS sequence diagram. Five turbo-flash readout trains are played repeatedly until all *k*-space is acquired. The first readout succeeds a T_2_-preparation module, and the remaining four readouts follow an inversion pulse that captures T_1_ dynamics. (**b**) Wave-MoDL diagram for multicontrast joint reconstruction. $A=WF_{y}PF_{x}C$ presents the forward model of wave-encoding. Convolutional regularizers are applied in both image- and *k*-space, which are combined with data consistency (DC) layers in an unrolled network structure. The model is trained in a supervised manner where the loss function measures the fidelity with respect to high-quality reconstructions at mild *R* = twofold acceleration.

**Figure 2 bioengineering-09-00736-f002:**
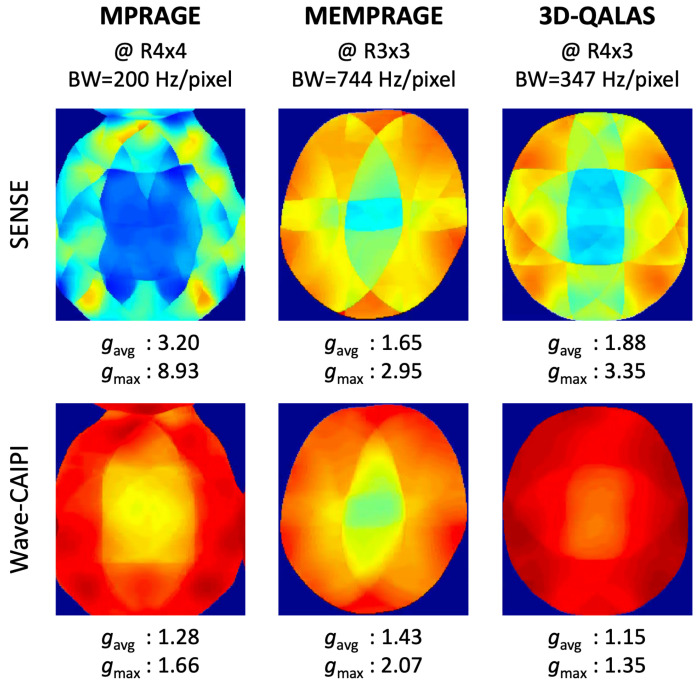
*g*-factor analyses of SENSE and wave-CAIPI at the target acceleration.

**Figure 3 bioengineering-09-00736-f003:**
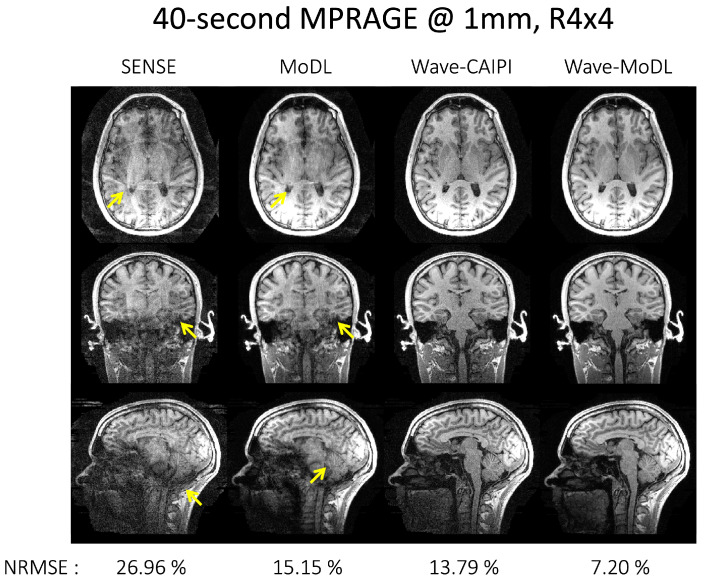
40 s 1 mm MPRAGE image reconstruction at *R* = 4 × 4. The arrows point to the folding artifacts and noise amplification.

**Figure 4 bioengineering-09-00736-f004:**
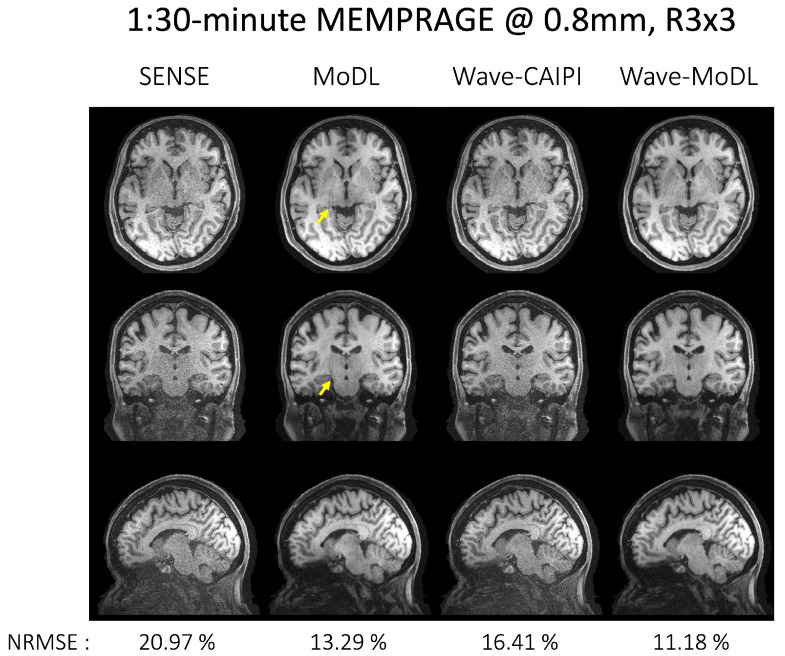
The proposed method on the MEMPRAGE database at *R* = 3 × 3-fold and 0.8 mm isotropic voxel resolution. Echo images were combined by calculating the root mean squared image. NRMSEs were calculated for the entire testing database.

**Figure 5 bioengineering-09-00736-f005:**
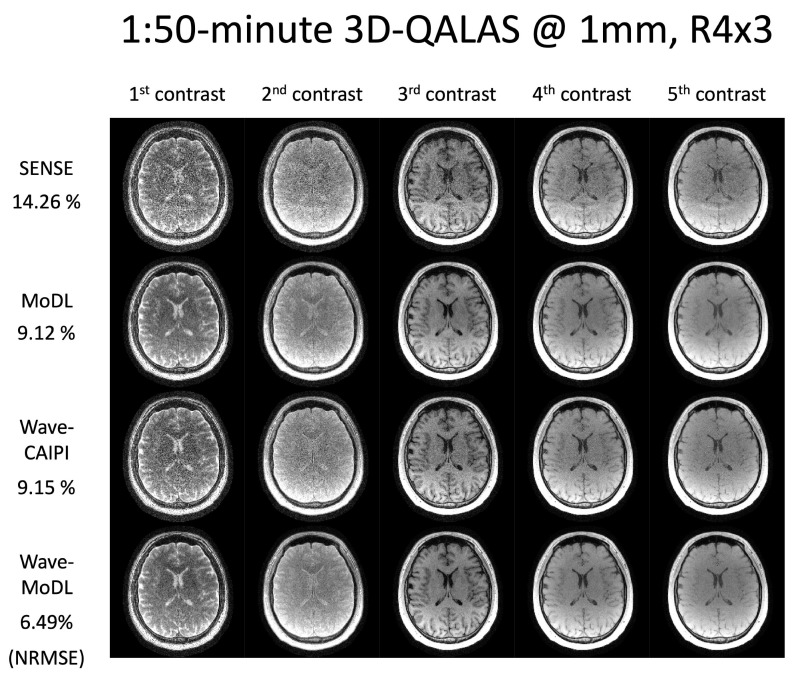
Multicontrast image reconstruction using SENSE, MoDL, wave-CAIPI, and wave-MoDL. The results are 1:50 min scans at *R* = 4 × 3 and 1 mm isotropic resolution.

**Figure 6 bioengineering-09-00736-f006:**
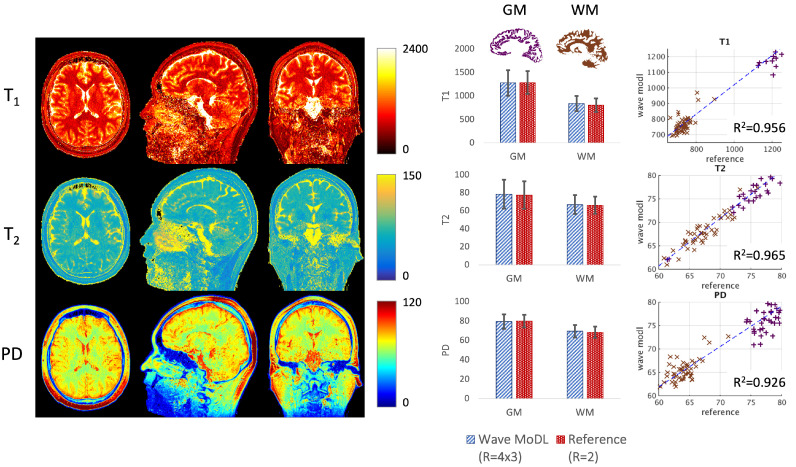
T_1_, T_2_, and PD maps were calculated using the wave-MoDL results. The last column shows the correction between the reference (*R* = 2) and the quantitative values of wave-MoDL (*R* = 4 × 3) in the randomly selected 5×5×5 boxes in gray matter (purple) and white matter (brown).

**Figure 7 bioengineering-09-00736-f007:**
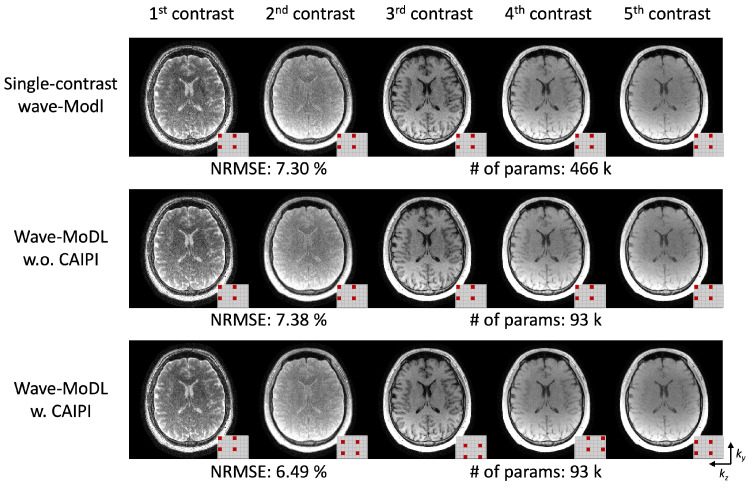
(**top**) Five individual single-contrast wave-MoDL reconstructions for each contrast and (**middle**, **bottom**) joint-wave-MoDL with and without CAIPI sampling pattern across contrasts. The small pattern in the right bottom corner of each image shows the sampling in the $k_{y}$ and $k_{z}$ directions.

**Figure 8 bioengineering-09-00736-f008:**
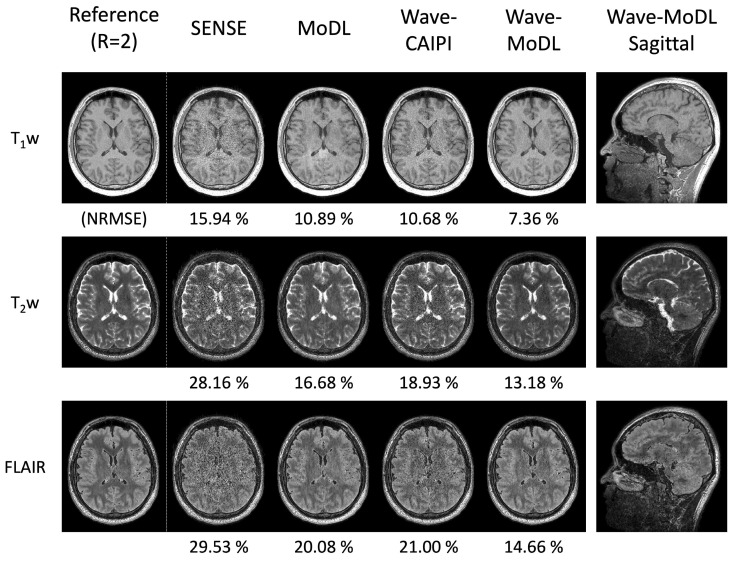
The synthesized T_1_w, T_2_w, and FLAIR images at *R* = 4 × 3-fold acceleration.

**Figure 9 bioengineering-09-00736-f009:**
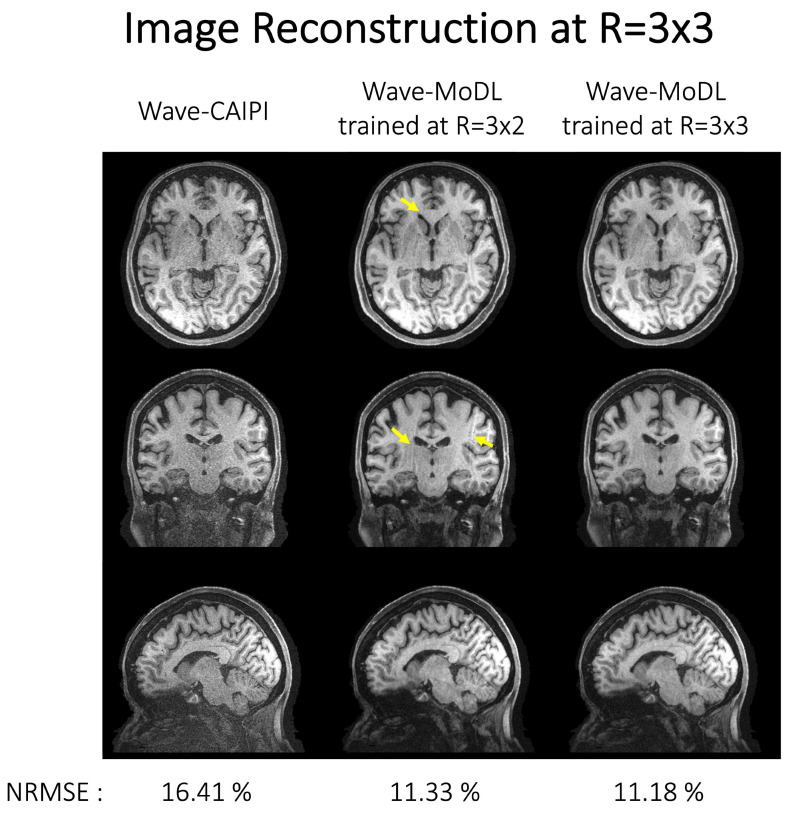
Image reconstruction on HCP database subsampled at *R* = 3 × 3 using (**left**) wave-CAIPI, (**middle**) wave-MoDL trained at *R* = 3 × 2, and (**right**) wave-MoDL trained at *R* = 3 × 3. NRMSEs were calculated for the entire testing database. The network would not be generalized and required to be fine-tuned at a different reduction factor or sampling pattern.

**Table 1 bioengineering-09-00736-t001:** Parameters of database and network parameters.

	MPRAGE	MEMPRAGE	QALAS
**Imaging plane**	Sagittal	Sagittal	Sagittal
**Voxel size** [mm^3^]	1×1×1	0.8×0.8×0.8	1×1×1
**FOV** [mm^3^]	256×256×192	320×300×208	240×238×192
**TR** [ms]	2500	2500	4500
**TI** [ms]	1100	1000	-/100/1000/1900/2700
**TE** [ms]	2.28	1.81/3.60/5.39/7.18	2.35
**Receiver bandwidth**	200 Hz/pixel	744 Hz/pixel	347 Hz/pixel
**Maximum wave gradient**	8.80 mT/m	9.63 mT/m	16.51 mT/m
**# of wave cycles**	11	4	5
**Acceleration**	4×4	3×3	4×3
**Scan time**	40 s	1 min 30 s	1 min 50 s
**#/depth of hidden layers**	5/24	5/24	5/24
**# of network parameters**	85,974	91,458	93,286
**# of subjects** (train/validate/test)	8/1/1	22/4/4	8/1/1

## Data Availability

The MPRAGE and 3D-QALAS data presented in this study are available on request from the corresponding author.

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
