# Peer review of "Wave-Encoded Model-Based Deep Learning for Highly Accelerated Imaging with Joint Reconstruction"

_bioengineering, 2022, doi:10.3390/bioengineering9120736_

Round 1
Reviewer 1 Report
Recently introduced model-based deep learning (MoDL) technique successfully incorporates convolutional neural network (CNN)-based regularizers into physics-based parallel imaging reconstruction using a small number of network parameters. Overall, this is an interesting study. I advise the writer to check the full text carefully, as there are still quite a few writing mistakes.
Author Response
Recently introduced model-based deep learning (MoDL) technique successfully incorporates convolutional neural network (CNN)-based regularizers into physics-based parallel imaging reconstruction using a small number of network parameters. Overall, this is an interesting study. I advise the writer to check the full text carefully, as there are still quite a few writing mistakes.
We thank the reviewer for the comment. We carefully corrected writing mistakes.
Reviewer 2 Report
Summary:
The paper considers extensive evaluation of Wave-CAIPI combined with MoDL. The authors show that it allows for higher acceleration factors, and can also be used to estimate T1, T2, PD parameters, hence allowing for contrast weighted images.
Strengths:
-The exposition is generally good.
- The study is extensive, with three datasets, and various accelerations, contrasts, etc.
- The T1, T2 parameter estimation seems strong.
- The folding artefacts and PSNR are noticeably better when compared to baselines.
Weaknesses:
- The methodological innovation is limited. The proposed algorithm is a straightforward combination of CAIPI and MoDL.
- This is related to the above weakness, but perhaps the authors can highlight what architectural / algorithmic changes were required for MoDL to work with CAIPI.
-The background in section 2 may not be the most helpful to unfamiliar readers. However, I do understand that the authors may have trimmed content there in order to save space, which is reasonable. Perhaps some references on why Wave-CAIPI allows for faster acquisition can be provided in section 2.
- MoDL is known to be a weaker algorithm when compared to several other deep learning models, such as U-Net. Why did you choose MoDL instead? The discussion sections has some details about U-Net but I didn't understand what the authors mean.
Notation / Clarifications:
- D_i and D_k in eqn 4 are undefined. I suppose they correspond to DC blocks in the pixel and k-space domain, but they should be defined in the appropriate sentence.
- How were the coil sensitivities estimated?
Author Response
We are thankful for the reviewer's comments. We have carefully addressed the concerns raised as below.
The methodological innovation is limited. The proposed algorithm is a straightforward combination of CAIPI and MoDL. This is related to the above weakness, but perhaps the authors can highlight what architectural / algorithmic changes were required for MoDL to work with CAIPI.
In this paper, we expanded the wave-CAIPI strategy to joint multi-contrast image reconstruction via a deep learning network, while different k-space sampling across multi-contrast has helped increase the collective k-space coverage. The DC layers enforced a wave-encoding forward model, which was made possible through external wave-PSF information. In addition, we reconstructed the 3D volume by separating the data into slice groups, thereby addressing GPU memory limitations that hamper processing volumetric k-space data.
L140: The data consistency (DC) block enforces consistency with measured wave-encoded signals where external wave-PSF estimation is required to utilize the known forward model.
L163: We separated the 3D data into slice groups and trained on sets of aliasing slices to address the GPU memory constraint. The number of slices of an input 3D batch image is equal to the acceleration factor in the slice-encoding direction. For example, at Rz =3, a batch contains 3 slices that are aliasing on each other slice and need to be unaliased. Multiple contrast images were concatenated in the input channel dimension of the network.
L334: We have shown that our proposed Wave-MoDL method enables highly accelerated MR scans that improve image fidelity and acquisition speed. Its application was demonstrated in MPRAGE, MEMPRAGE, and 3D-QALAS sequences. Joint wave-MoDL was introduced to improve the reconstruction by employing the information from other echoes/contrasts using CAIPI sampling patterns.
The background in section 2 may not be the most helpful to unfamiliar readers. However, I do understand that the authors may have trimmed content there in order to save space, which is reasonable. Perhaps some references on why Wave-CAIPI allows for faster acquisition can be provided in section 2.
We added more references and explanations to clarify the utility of wave encoding.
L106: Wave-encoding employs additional sinusoidal gradients during the readout to take better advantage of 3D coil sensitivity profiles [7,8,10,43]. Wave-encoding modulates the phase during the readout and incurs a corkscrew trajectory in k-space. This spreads the aliased signal into the readout direction, thereby employing an additional sensitivity profile for image reconstruction.
MoDL is known to be a weaker algorithm when compared to several other deep learning models, such as U-Net. Why did you choose MoDL instead? The discussion sections has some details about U-Net but I didn't understand what the authors mean.
MoDL incorporates physics-based image reconstruction utilizing the known forward model. This will help guide the reconstruction using the acquired data and enforce consistency to the known physics-based model. In addition, MoDL allows us to reduce the number of network parameters. This helps train the network with a relatively small database while preserving the performance (Ref [32]).
L64: The sharing of network parameters across iterations enables MoDL to keep the number of learned parameters decoupled from the number of iterations, thus providing good convergence without increasing the number of trainable parameters. Such smaller number of trainable parameters also translates to significantly reduced training data demands, which is particularly attractive for data-scarce medical-imaging applications.
D_i and D_k in eqn 4 are undefined. I suppose they correspond to DC blocks in the pixel and k-space domain, but they should be defined in the appropriate sentence.
We added the explanation of Dk and Di as follows.
L146: By substituting ηn+1 = Dk(mn+1) and ζn+1 = Di (mn+1), where Dk and Di represent CNNs in the k- and image-space, respectively, an alternating minimization-based solution can be described as follows.
How were the coil sensitivities estimated?
We explained how we estimated the coil sensitivity maps in the revised manuscript.
L172: Coil sensitivity maps were calculated from external 3D low-resolution gradient-echo-based reference scans using ESPIRiT.
Reviewer 3 Report
1) The language is very technical. Someone, who would like to apply the approach for obtaining better results in medical imaging would benefit from a short description of the method, findings, and how to apply it in practice in simple terms. A couple of paragraphs can be included in the introduction or in the conclusion, so that a non-specialist in the narrow field can quickly understand the essence of the paper and how to use the approach.
2) The authors state (lines 164-166) “We acquired fully sampled T1-MPRAGE data on 10 healthy subjects at 1-mm isotropic resolution to generate the database. We used a split of 8/1/1 subjects for training /validating /testing the wave-MoDL network.” Using such small samples does not seem convincing. If the authors could not use larger samples, they have to explain why it is so and why the results can still be valid.
3) Some small technical problems:
a. Figures 1 and 6 are too small. The text is not legible and some small details are indistinguishable.
b. Line 264: images[41] -> images [41] (there should be space before the opening bracket).
Author Response
We are thankful for the reviewer's comments. We have carefully addressed the concerns raised as below.
1) The language is very technical. Someone, who would like to apply the approach for obtaining better results in medical imaging would benefit from a short description of the method, findings, and how to apply it in practice in simple terms. A couple of paragraphs can be included in the introduction or in the conclusion, so that a non-specialist in the narrow field can quickly understand the essence of the paper and how to use the approach.
We provided more information that can help describe the essence of the paper in the revised Introduction section.
L20: MRI has been widely used to provide structural and physiological images, however, its efficiency has been limited by an inherent tradeoff between scan time, resolution, and signal-to-noise ratio (SNR) [1]. MRI repeatedly excites and collects the signal while encoding the signal to visualize the acquired data as images, which often takes a long scan time.
L75: Quantitative MRI estimates tissue parameters quantitatively, enabling the detection of e.g. microstructural processes related to tissue remodeling in aging and neurological diseases [34].
2) The authors state (lines 164-166) “We acquired fully sampled T1-MPRAGE data on 10 healthy subjects at 1-mm isotropic resolution to generate the database. We used a split of 8/1/1 subjects for training /validating /testing the wave-MoDL network.” Using such small samples does not seem convincing. If the authors could not use larger samples, they have to explain why it is so and why the results can still be valid.
Because the network was trained slice-group-by-slice-group, the network had access to 384 slice groups during the training. MoDL shares the weights for every iteration so it consists of a relatively small number of network parameters. MoDL MRI paper, Ref [32], used 4/1 subjects to train/test the network. They selected 360 slices to train the data, and verified that their network consisting of 188K parameters was sufficiently trained with their database. Our network consists of 86K parameters, as such, we anticipate that 388 slice groups should be sufficient for training.
L178: The network had access to 384 slice groups during the training, which should be a sufficient database size considering the number of network parameters as detailed in [32].
3) Some small technical problems:
Figures 1 and 6 are too small. The text is not legible and some small details are indistinguishable.
Line 264: images[41] -> images [41] (there should be space before the opening bracket).
We enlarged the figures and text and carefully corrected technical problems.